# Urban Hedgehog Behavioural Responses to Temporary Habitat Disturbance versus Permanent Fragmentation

**DOI:** 10.3390/ani10112109

**Published:** 2020-11-13

**Authors:** Anne Berger, Leon M. F. Barthel, Wanja Rast, Heribert Hofer, Pierre Gras

**Affiliations:** 1Department of Evolutionary Ecology, Leibniz Institute for Zoo and Wildlife Research, Alfred Kowalke Straße 17, 10315 Berlin, Germany; Barthel.leon@gmx.de (L.M.F.B.); rast@izw-berlin.de (W.R.); 2Berlin Brandenburg Institute of Advanced Biodiversity Research (BBIB), 14195 Berlin, Germany; hofer@izw-berlin.de (H.H.); pierregras@gmx.de (P.G.); 3Department of Biology, Chemistry, Pharmacy, Freie Universität Berlin, Takustrasse 3, 14195 Berlin, Germany; 4Department of Veterinary Medicine, Freie Universität Berlin, Oertzenweg 19b, 14195 Berlin, Germany; 5Department of Ecological Dynamics, Leibniz Institute for Zoo and Wildlife Research, Alfred Kowalke Straße 17, 10315 Berlin, Germany

**Keywords:** disturbance, fragmentation, anthropogenic habitat change, urban ecology, behavioural plasticity, GPS telemetry, hedgehogs

## Abstract

**Simple Summary:**

Wildlife is exposed to environmental disturbances. Some are limited to a short period and pass by, others are of a permanent nature. Often these two kinds of disturbances occur simultaneously. This makes it difficult to disentangle the specific behavioural response to each disturbance. As species may respond to different disturbances in different ways, it is important to know the species-specific and disturbance-specific responses to develop effective species conservation action. We investigated the behavioural responses of European hedgehogs (*Erinaceus europaeus*) in Berlin to temporary disturbance (in the form of an open-air music festival) and permanent disturbance (in the form of habitat fragmentation). We show that a music festival is a major stressor that strongly influences all investigated behaviours. Urban hedgehogs in a highly fragmented area showed subtle behavioural changes compared to those in low-fragmented areas, suggesting that fragmentation was a moderate challenge which they could cope with. Thus, the temporary disturbance by a music festival had a more serious impact on hedgehog behaviour than permanent disturbance caused by fragmentation. Moreover, we show that males responded stronger to the transient disturbance and females responded stronger to habitat fragmentation.

**Abstract:**

Anthropogenic activities can result in both transient and permanent changes in the environment. We studied spatial and temporal behavioural responses of European hedgehogs (*Erinaceus europaeus*) to a transient (open-air music festival) and a permanent (highly fragmented area) disturbance in the city of Berlin, Germany. Activity, foraging and movement patterns were observed in two distinct areas in 2016 and 2017 using a “Before & After“ and “Control & Impact“ study design. Confronted with a music festival, hedgehogs substantially changed their movement behaviour and nesting patterns and decreased the rhythmic synchronization (DFC) of their activity patterns with the environment. These findings suggest that a music festival is a substantial stressor influencing the trade-off between foraging and risk avoidance. Hedgehogs in a highly fragmented area used larger home ranges and moved faster than in low-fragmented and low-disturbed areas. They also showed behaviours and high DFCs similar to individuals in low-fragmented, low disturbed environment, suggesting that fragmentation posed a moderate challenge which they could accommodate. The acute but transient disturbance of a music festival, therefore, had more substantial and severe behavioural effects than the permanent disturbance through fragmentation. Our results are relevant for the welfare and conservation measure of urban wildlife and highlight the importance of allowing wildlife to avoid urban music festivals by facilitating avoidance behaviours.

## 1. Introduction

A disturbance describes a change in an environment that poses a change in the ecosystem [1]. Nowadays, all ecosystems are increasingly subjected to anthropogenic disturbances which include a host of features, from artificial light, noise and air pollution to habitat fragmentation [2,3,4]. When habitats are disturbed, animals must reconsider trade-offs between foraging or mating success and risk aversion similar to the landscape of fear [5,6]. Humans may change these environments slowly, e.g., through changes over decades, or fast within a few days when building a road, removing a forest patch or harvesting hay from a meadow [4,7,8]. Fast changes require individuals to show appropriate behavioural flexibility or plasticity within their own lifetime to cope. If facing frequent or regular disturbances, such flexibility could contribute to population viability. On the individual scale, there are several options to respond to disturbance: disperse and seek another place to live or stay or adjust through behavioural plasticity [4,6]. On the population scale, genetic changes may be possible to adapt to the new environmental conditions if they are permanent and exert sufficient selection pressure [7].

It can be challenging to disentangle the effects of transient and permanent stressors as different forms of disturbance often operate simultaneously. In this study, we investigate the effects of habitat fragmentation and transient disturbance on individual spatiotemporal behavioural responses in urban European hedgehogs (*Erinaceus europaeus*) by monitoring two populations in Berlin, Germany. One population lives in a habitat with permanent fragmentation, the other faced several severe transient disturbances associated with a music festival.

Festivals often take place on open green areas. Within days, these green areas will be modified and disturbed through heavy machinery and the presence of many people as the festival’s infrastructure is being set up. During the festival, a huge crowd of people wanders around the festival site, loud music plays throughout most of day and night and for enjoyment and safety, the whole area is lit up and fenced. Thus, during the festival, wildlife living on the site is confronted with extraordinary amounts of habitat changes, noise, light pollution and people activities.

Fragmentation is created by transforming habitats into smaller patches, thereby creating a permanent mosaic-like landscape [9,10,11]. Thus, fragmentation can impede movements in the matrix between resource patches [12] and limit access to mating partners [13,14], forcing animals to adjust their behaviour to these changes [15]. Some examples of behavioural plasticity include increasing home ranges and adjusted activity rhythms to cope with fragmentation [16,17,18].

The European hedgehog (*Erinaceus europaeus*) is a solitary, hibernating, nocturnal insectivore which lives in a wide range of habitat types [19,20]. Long-term monitoring studies found that the overall hedgehog population in several countries is declining, sometimes dramatically so [21,22,23,24]. The underlying mechanisms responsible for these declines are likely to be complex and multifactorial, including habitat loss, interspecific competition, traffic accidents and intensification of agriculture [21,25,26,27,28]. Increasing fragmentation and decreasing hedgehog densities may cause negative genetic effects associated with isolated populations [29,30,31,32]. Usually, urban green spaces such as public parks are favourable environments for urban hedgehogs, providing easily accessible food and resting (nest) sites. If the urban matrix provides a very patchy environment with only small, isolated patches of suitable habitat dispersal out of unfavourable habitats can be problematic and limited [7]. It is at present unclear how hedgehogs cope with habitat fragmentation as an example of environmental disturbance and how this might influence their population structure. In order to develop effective conservation concepts, threats to hedgehogs as well as their limits of adaptability should be known. Due to their cryptic lifestyle, little is known about their behaviour in the wild, the effect of different disturbances on their behaviour, and the limits to hedgehog adaptability [28,33].

In the case of habitat fragmentation, hedgehogs need to trade-off between foraging profitability and safety, by changing the distance to anthropogenic sources of disturbance, and reconsider options for easily accessible food. To investigate the behavioural plasticity of hedgehogs in response to transient changes and habitat fragmentation, we analysed the behavioural response at different levels. First, we used GPS data to investigate the daily home range size and movement behaviour of hedgehogs, both closely related to foraging [34,35]. Second, we assessed the circadian behaviour patterns of hedgehogs, known to be usually strictly nocturnal but adjustable as a response to stress [19,35]. Third, we monitored nesting behaviour. Nests are important resting sites of both sexes and typically used for several days [36,37], females also keep their offspring in nests.

We investigated two scenarios. First, we studied effects of a transient and intense disturbance in the Treptower Park in Berlin, Germany, which has little habitat fragmentation and low levels of night-time disturbance outside the festival period. Second, we investigated the effects of habitat fragmentation in the Tierpark Berlin, which has many daytime visitors but very low nightly disturbance. We hypothesise that both transient disturbance and habitat fragmentation influence space use, activity patterns and nesting behaviour to different degrees.

We predict the following behavioural adjustments in response to (A) transient disturbance:(1)Regarding space use, we predict that hedgehogs avoid transient disturbed areas in their habitat, by either leaving the area or adjusting movement patterns: (1a) We predict avoidance of the disturbed (festival) area by shifting the centre of the nightly used area. (1b) Additionally, we predict a decreased size of nightly home ranges in or close to the festival area due to avoidance of the disturbed (festival) area.(2)We predict that hedgehogs adjust their movement behaviour. The animals now have to look for the same amount of food in a potentially less favourable and/or smaller area and thus foraging effort may have to be increased. We, therefore, predict an increase in search intensity, greater turning angles and slower speed under disturbance.(3)The general levels of activity will be reduced due to increased vigilance behaviour which in hedgehogs is characterised by immobility (little activity).(4)We predict that high levels of disturbance during the festival induces females and males to switch their nests more often and the number of days spent in the same nest decreases.

We predict the following behavioural adjustments in response to (B) fragmentation:(1)In urban areas, or areas with predation risk, hedgehog movements are strongly associated with linear structures [35], fragmentation will increase the area of space that is of no interest to hedgehogs and thus increase the distances they have to cover. Thus, in a highly fragmented park, the home range area would be bigger than in the low-fragmented park.(2)As fragmentation is likely to increase distances for commuting between favourable food patches, movement characteristics should change. We expect a faster speed, a larger number of smaller turning angels and a lower search intensity than in a low-disturbed, low-fragmented habitat.(3)Fragmentation could influence nesting behaviour in two ways: animals either have to change their nests more frequently to be closer to favourable food patches, or extend their stay in nests if they are close to favourable food patches. During disturbances, we predict a more frequent nest change than in undisturbed areas or at undisturbed time.

## 2. Materials and Methods

### 2.1. Study Areas and Study Design

Fieldwork was conducted between 10th of August and 20th of September 2016 in the Treptower Park, in southeast Berlin, Germany (52.48846° N, 13.46974° E) and between 14th of August and 4th of September in 2017 in the Tierpark, a big park containing a zoological garden in East Berlin, Germany (52.50326° N, 13.52976° E). Both parks include big trees and several green spaces of short grass swards with shrubs of various sizes and hedges. Additionally, there are playgrounds, larger sealed areas and footpaths. Treptower Park is a public city park of 88.2 ha in size; it is open to the general public 24 h and 7 days per week whereas the Tierpark is closed to the general public from dusk to the morning. The Tierpark is a zoological garden of about 160 ha in size; it contains numerous animal enclosures, small buildings, water ditches and many concrete footpaths, creating a mosaic-like fragmented habitat with many areas that are non-accessible to hedgehogs. The maintenance of the parks is similar, with leaf litter being removed from some areas, particularly the footpaths, but left in bushes and scrub throughout the year, offering a similar habitat in both parks suitable for hedgehogs and other wildlife.

In 2016, the Lollapalooza Festival with over 140,000 visitors took place in the Treptower Park for which a substantial portion of the park (excluding a war memorial and the southeastern segment) was temporally substantially changed. Music stages, amusement facilities and enclosures were constructed and built between 29th of August and 9th of September, the festival took place on 10th and 11th of September, and deconstruction of all facilities took place from 12th of September onwards. We collected data on hedgehog movements and behaviour before the festival (pre-festival) until construction work for the festival started and during the festival-phase, including the time periods of construction and deconstruction. The festival phase lasted from 29th of August until 16th of September 2016. The pre-festival period is defined from 10th of August until 28th of August and represents the control for both the transient disturbance caused by the festival and, as an example of hedgehogs living in a low-fragmented and low disturbed urban habitat, for the hedgehogs studied in the Tierpark.

To investigate the effect of the transient disturbance, all data from hedgehogs in the Treptower Park were used by comparing those collected before and during the festival. To investigate the effect of fragmentation change, data from the highly fragmented park (Tierpark) were compared with the pre-festival data from the low-fragmented park (Treptower Park).

### 2.2. Hedgehog Capture and Logger Attachment

At the beginning of each study period, surveys were carried out during two to three nights at least one hour after sunset to find active hedgehogs by spotlighting (P14.2, LED Lenser, Solingen, Germany). Every hedgehog located was marked with five shrink tubes glued to the spines [38]. The tubes were labelled with a number starting with 1 to identify them during recapture [39]. From all previously captured hedgehogs, we selected eight hedgehogs (four of each sex) and equipped them with Global Positioning System (GPS)/Accelerometer (ACC) loggers (e-obs GmbH, München, Germany) and Very High Frequency (VHF) (Dessau Telemetrie, Dessau, Germany) transmitters using a backplate system [40]. We only used hedgehogs with a body mass exceeding 600 g to ensure that the attached logger equipment was below the 5% body mass rule [41,42].

During the study, all hedgehogs were weighted and inspected for any problems once a week; these occasions were also used for the necessary recharging of the GPS/ACC data loggers. At the end of the studies, the animals were caught again, weighed, the loggers removed, and the backplate system was cut off the spines.

All procedures performed involving the handling of animals were in accordance with the ethical standards of the institution (IZW permit 2016-02-01) and German federal law (permission number Reg0115/15 and G0104/14).

### 2.3. Logger Sampling Setup

In order to find the animals with data loggers at any time, VHF transmitters continuously broadcast signals throughout the whole study period. GPS positions were taken during expected activity times of hedgehogs from 7:00 p.m. to 7:00 a.m. in 10 min intervals, in bursts of five points, which means that the GPS record 5 singular GPS positions of one-second difference. Acceleration data were recorded alongside GPS data every minute. These three-dimensional accelerometers were programmed to record a short burst of simultaneous high-resolution data. A sampling frequency of 100 Hz per axis was chosen for the present study. A burst took 2.64 s for two individuals (01_2016 and 19_2016); the other individuals were recorded with bursts of 2.5 s. This difference in burst length is not ideal, although the burst length is only important for three out of 25 predictors used for the model (see “Behaviour prediction and budget” in Section 2.5.2).

### 2.4. Nesting Behaviour Monitoring

Nesting behaviour was recorded every day by locating the VHF signals of each hedgehog carrying a logger (TRX-1000S, Wildlife Materials Inc., Murphysboro, IL, USA, or Wide Range Receiver AR 8200, AOR Ltd., Tokyo, Japan). The nest position was recorded using a Garmin GPSmap 60CSx device (Garmin Deutschland GmbH, Garching, Germany) every day for the nest the hedgehogs stayed in during the daytime. If a hedgehog was found in the vicinity (2 m) of a nest without a new nest, the existing nest was noted as the day nest of the hedgehog. Some animals had to be removed from the dataset of 2017 in the Tierpark because they occupied fewer than 5 nests. If a hedgehog lost a transmitter and could not be found to re-attach the transmitter, or night work had been discontinued, the nest surveys for the individual were stopped.

### 2.5. Analyses

To investigate the effect of the transient disturbance of the festival, all data (GPS, ACC and nest behaviour) from the hedgehogs in the Treptower Park were used by comparing before and during the festival as a temporal control. To investigate the effect of permanent habitat change (fragmentation), data from the highly fragmented park (Tierpark) were compared with data from the low-fragmented park (Treptower Park) from the pre-festival period in a spatial control.

#### 2.5.1. GPS Data

Because of high fluctuations in some GPS points, we excluded all points that were more than 1000 m away from the study site. We then calculated the average of all remaining GPS points measured within a burst and we calculated the distance between two consecutive GPS positions to derive the speed the hedgehogs had to overcome for that distance. Outliers of more than 2 m/s speed from one location to the next were excluded. Overall, the mean error of the GPS position was between 10 and 40 m.

To conserve the natural variability, we decided to use every night as a single event and calculated the following values accordingly using R and Rstudio [43,44]. To assess the nightly used area, we calculated the 95% Minimum Convex Polygon (mcp95) and the Kernel density estimates (kde50) as a core area of use by smoothing with the ad hoc method (href) [45]. In both cases, we used a linear model [46] to perform a linear mixed effects analysis on the relationship between used area (mcp95 or kde50) and our three treatments. As fixed effects, we entered treatment and sex with interaction into the model. As random effects, we had intercepts for individuals. We did not include age as a covariate because age determination in hedgehogs is generally difficult. Moreover, we can guarantee that all hedgehogs were adults, meaning that they were born at least a year before the study To fit these assumptions, we included a power function using varPower() as weights. Visual inspection of residual plots did not reveal any obvious deviations from homoscedasticity or normality. To obtain *p*-values of the mixed-effect model, an ANOVA (Type II) was performed on the fitted model using the ‘Anova’ function from R-package ‘car’ and applying a Wald chi-square tests [47]. Subsequently, differences between groups of the fixed-effects of the fitted model were tested using the R package ‘multcomp’ [48]. We compared the groups: pre vs. pre comparing both sexes, pre vs. festival within sex, pre vs. fragmented within sex, fragmented vs. fragmented comparing both sexes.

We proceeded in a similar manner when analysing the movement speed of hedgehogs as a travelled distance for a time interval between two consecutive GPS positions (m/s). We used the lme4 package to perform a linear mixed effects analysis on the relationship between speed and treatment [49]. As fixed effects, we entered treatment and sex (with interaction) and as random effects, we had intercepts for individuals. Visual inspection of residual plots did not reveal any obvious deviations from homoscedasticity or normality. *p*-values were obtained by applying an ANOVA (Type II) and group comparison as described above.

To evaluate how animals use the available habitat, we calculated a ratio of area used (mcp95 in (m^2^)) and distance travel (m) per night (calculated with st_length [50]), resulting in a measure of search intensity with units (m/(m^2^ × d)) or moved distance per square metre and day. To evaluate whether treatment or sex had an effect on this parameter, we used the SpaMM package [51] by first finding the right fit and then comparing the null model with different models. The SpaMM package was necessary to counter auto-correlation.

To detect wherever the hedgehogs shifted their utilised area during the festival, the longitude and latitude values of centroids per night were used separately using function centroid() [52]. We normalised the values by subtracting the mean value from the pre-festival phase and worked with the absolute values. For the latitude and longitude values, linear mixed effect models were fitted with treatment as the fixed effect and sex as the random effect. Both time values had to be square-root transformed before fitting the model to meet the assumptions of homoscedasticity or normality. P-values were obtained from ANOVA (Type II) as described above.

Movement of hedgehogs was further characterized by calculating turning angles [53] and plotted as absolute values because we were interested in the general movement. Results were then randomly sampled and compared in a permutation approach 1000 times using a two-sample two-sided Kolmogorov–Smirnov test [43]. This comparison was only done on the treatment level (pre-festival vs. festival and pre-festival vs. fragmentation).

#### 2.5.2. ACC/Acceleration Data

To account for missing data because of recharging or logger malfunctioning, all data with fewer than 1430 measurements between 00:00 and 23:59 were removed from the data set (*n* = 1440 for complete 24 h). This removal of data ensured that only days with a comparable length and the same number of records during days and nights were considered for the analysis and, therefore, did not favour behaviours that only occurred during a specific time of the day.

##### Behaviour Prediction and Budget

Acceleration raw data were tested for missing measurements within the bursts. All bursts where fewer data were recorded than intended by the settings (*n* = 264) were removed. We sued the remaining data for behaviour detection by applying a supervised machine learning algorithm. The train and test dataset for the behaviour recognition were taken from a previous study. The whole procedure is described in [54]. By joining multiple Support Vector Machines (SVM), the selected behaviours were classified. Here, we considered three behaviour classes: resting, balling up and locomotion (referred to as walking). To account for behaviours that are not included in the model but might occur in hedgehogs, a threshold for the probability belonging to a class of 0.7 was set for the SVM. Otherwise, the behaviour was classified as “other” behaviour.

The SVM model was then used to assign a predicted behaviour to every burst and its corresponding timestamp. The behaviour of every individual was treated for the following tests separately. To test for effects on behaviour classes, a general linear model was performed [49] taking a quotient of the behaviour in relation to all behaviours. As a fixed effect, the treatment, as well as the sex and the interaction, were put into the model. Individuals were included as random effect. This was followed by an analysis of variance [47] and general linear hypotheses and multiple comparisons [48] using the same matrix as before.

##### Stress Detection via the Degree of Functional Coupling (DFC)

The Degree of Functional Coupling (DFC) is a measure for the synchrony of (internal) cyclic behaviour and the (external) environmental 24 h rhythm [55,56]. To calculate DFCs, the standard deviations of raw acceleration data of all three axes were calculated and summed up per measurement interval (burst). Following the protocol of Berger et al. [56,57], this time series was autocorrelated in order to filter out the noise and enhance rhythmic components. Afterwards, a Fourier transformation was used to break it down into its rhythmic components, as described by the percentage of each component in the original time series. The longest Fourier period tested covered the entire length of the autocorrelation function (here three days); the shortest Fourier period tested was twice the sampling interval (here 2 min). The DFC is then calculated by dividing the portion of Fourier transformation components that harmonize with the 24 h rhythm by the entirety of the Fourier spectrum. To gain an adequate statistical power of the 24 h period, DFC were calculated for time series of three days equivalent to the procedure of a moving average (first data set covers day 1 to 3, second data set covers day 2 to 4 and so on). The resulting DFCs were assigned to the day of the three days that entered the calculation for the first time. These data were then analysed using a linear mixed effect model with treatment and sex and their interaction. Values had to be arcsine transformed in order to meet the assumptions of homoscedasticity and normality. Afterwards, an analysis of variance [47] and general linear hypotheses and multiple comparisons [48] were performed.

#### 2.5.3. Nesting Behaviour

For each nest, the duration of occupation was scored as exact if both starting and stopping dates of nest use were recognised, or as right-censored (a minimum estimate), if either the starting date or the stopping date at the beginning and end of study periods were not known. Then, the survivorship function was calculated using package survival [58] separately for both parks and treatment conditions. If significance was found, a post hoc Mantel test was performed to detect the source of the difference.

## 3. Results

Data were collected for 16 hedgehogs (8 of each sex) over a period of 3 to 41 nights per animal; between 95 and 2566 GPS locations per animal were recorded (Appendix A). In total, 39 days from the Tierpark data set and 79 days from the Treptower Park data set were removed from further analyses due to high number of outliers or missing data. In total, we tracked 426 nights including 156 nights for the control, 152 for the festival and 118 for the highly fragmented site. Sexes are represented with 236 females and 190 male data points. In Appendix A, we mapped GPS locations of some hedgehogs in the two parks to indicate movement characteristics (paths) and spatial obstacles for the hedgehogs which they have to run around.

The daily maximum temperatures were between 18 and 34.5 °C (mean = 25.2 °C) during the 2016 study period (41 days) and between 18 and 30 °C (mean = 22.1 °C) during the 2017 study period (21 days). The lowest nighttime temperatures ranged between 7 and 18.4 °C (mean = 13.8 °C) during the 2016 study period and between 6.3 and 21 °C (mean = 13 °C) during the 2017 study period. The mean rainfall was 0.8 mm/day (11 days with rain) during the 2016 study period and 1.1 mm/day (10 days with rain) during the 2017 study period (https://www.wetteronline.de/, further weather information is listed in Appendix A).

### 3.1. Home Range Size

The nightly used area (Figure 1a) measured by the mcp95 was significantly smaller during the festival (χ^2^ = 54.82, df = 2, *p* < 0.001) and in females (χ^2^ = 6.48, df = 1, *p* = 0.011, details on all models Appendix A). Although treatment was similar between sexes (χ^2^ = 1.74, df = 2, *p* = 0.42), females in the control group used smaller areas by 1.9 times than males in the control group (2.55 ha to 4.71 ha, respectively). While both sexes decreased their home ranges during the festival (females acute change estimate E = −1.029 ± 0.22 (standard error), z = −4.648, *p* < 0.001; males −1.467 ± 0.38, z = −3.890, *p* < 0.001), only females increased the area in the highly fragmented habitat (E = 1.942 ± 0.69, z = 2.796, *p* = 0.028). Male hedgehogs occupied a bigger area than females in the control group and showed only a slight increase in the highly fragmented area from the control group (E = 0.730 ± 0.8663, z = 0.843, *p* = 0.92). Thus, males and females in the highly fragmented habitats had similar-sized home ranges (E = 0.6849 ± 0.8706, z = 0.787, *p* = 0.9390). These were replicated by the used core area measured by the kde50, except for the comparison between pre-festival and fragmented within the females (Appendix A).

### 3.2. Movement Speed and Turning Angle

Hedgehogs moved significantly faster in the highly fragmented area (0.049 m/s ± 0.001) than in the low-fragmented area during the control period (0.040 m/s ± 0.008, Figure 1b; χ = 33.3, df = 2, *p* < 0.001). During the festival, hedgehogs moved even more slowly (0.038 m/s ± 0.008, Figure 1b). Mean search intensity (m/m^2^ × d) was lowest for the control group and the highest during the festival (χ^2^ = 7.42, df = 2, *p* = 0.024); the highly fragmented area has the largest confidence interval in search intensity (Figure 1c). There was no difference between the sexes (χ^2^ = 0.11, df = 1, *p* = 0.74) nor was the interaction significant χ^2^ = 3.22, df = 2, *p* = 0.20). Although the curves of the subsample of the turning angles appear to look different, there was no significant difference in the characteristics of the turning angles (Figure 1d).

### 3.3. The Centre of the Nightly Home Range

The centroid values of the daily used area present the mean point of the used area and should have shifted if the hedgehogs used other areas, and the distribution should change if they avoid certain areas. During the control period, hedgehogs focused on a central big open meadow, both sexes had a near to normal distribution in their longitudinal values (Figure 2). During the festival period, hedgehogs moved significantly away from their mean centroids of the control period by on average ~35/30 m (longitudinal/latitudinal) in females and more than 65/105 m in males (effect of treatment latitude χ^2^ = 80.59, *p* < 0.001/longitude χ^2^ = 80.86, *p* < 0.001, with the interaction of treatment and sex χ^2^ = 21.44, *p* < 0.001/χ^2^ = 11.79, *p* < 0.001).

### 3.4. Behaviour Parameters

There were significant differences in the behaviours between the control (pre-festival) and festival period (Figure 3). In females and males, forming a ball (balling behaviour) was detected more frequently during the festival than the control period, females showed an increase of 0.153 and males of 0.2 (Figure 3a). For walking behaviour, only males showed a significant increase of 0.03 (Figure 3b). Resting behaviour was identified less in both sexes during the festival (−0.21 in females, −0.014 in males, Figure 3c). In the DFC values as an indicator of the degree of synchrony of the overall behaviours with diurnal rhythm, both sexes showed the same patterns. Both sexes showed similar values in the control group, whereas the values during the festival were reduced. In the highly fragmented area, we found high DFC values for females and slightly lower DFC values with high variation for males (Figure 3d).

### 3.5. Nesting Behaviour

In the control group, female hedgehogs used their nests again on the next day in 66.1% of cases. Nests of male hedgehogs were used with a probability of 57.8% on the next day. The comparison between the control and the festival period showed significant changes in nesting behaviour and differed between the sexes: During the festival, nests of male hedgehogs were used over significantly shorter time periods (Mantel logrank test, N = 156, *p* = 0.02) and the probability of using a new nest the next day was reduced from 57.8% to 45.5%. After eight days at the latest, males had left the nest and moved to another one. In contrast, for females, values were in general similar to or higher than during the control period (Mantel logrank test, N = 88, *p* = 0.83, Figure 4).

## 4. Discussion

Consistent with our predictions, there was an influence of fragmentation as well as of transient changes in habitat on hedgehog spatiotemporal behaviours. However, hedgehogs demonstrated sex-specific responses to different types of disturbances.

By using the same measuring method, the same number of animals, the same gender ratio, the same season, similar park sizes (in parts of the city with comparable urbanity index) and hedgehog population densities, we tried to achieve the greatest possible comparability between the two study areas and years. The two study areas, Treptower Park and Tierpark, are comparable considering their sealing index [59], which closely corresponds to other urbanization indices, such as human population density, disturbance by humans and pets, noise and light pollution [60,61]. In hedgehogs with main natural prey being earthworms and ground beetles, the sealing index will also be linked to food availability.

Compared to Treptower Park, the Tierpark is clearly more fragmented for hedgehogs, which is also the most obvious difference between the two parks: in Treptower Park almost all fences can be slipped through by hedgehogs. Only a long semicircular wall around the Soviet memorial in the middle of the park is impassable for hedgehogs. The Tierpark, on the other hand, consists of a large number of enclosures, most with borders insurmountable for hedgehogs (moats with wall edges, dense fences as protection against rats). Even if the hedgehogs manage to get into the enclosures, it can be more dangerous for them there than outside—for example, keepers once found a pregnant hedgehog kicked to death by takins (*Budorcas bedfordi*) in an enclosure. Hedgehogs therefore usually move around the enclosures, often on visitor paths (see Appendix A, male 2017_31).

Moreover, the weather data of the two study years do not differ significantly from each other: both summers were relatively warm; at least warmer than the average temperatures of 4 reference years (Appendix A). Summer 2016 was dryer than average, in 2017 rainfall was above average. The average temperatures of September 2017 were lower than those of September 2016, and the amount of rainfall in August 2017 was higher than in August 2016. However, the variance of weather data during both study periods was greater than the differences between the two. As a warmer drier summer will make earthworms harder to find and, therefore, hedgehogs need to roam further to meet their energy requirements, then hedgehogs in our study should have had smaller home ranges in 2017 than in 2016. However, our measurements on hedgehogs in the Tierpark in 2017 and in 2018 (unpublished data) demonstrated much larger distances and home ranges than those of the hedgehogs in Treptower Park.

Studies on urban foxes also showed that seasonal differences (and thus differences between years) are attenuated due to the permanent availability of food [62]. In this respect, we have come to the conclusion that the slightly different weather conditions in the two study years, although they are boundary conditions to be considered, are not causally relevant for our study results.

Due to transient changes of the festival, hedgehogs decreased home range size, movement speed and rested less but increased search intensity, performed balling up behaviour more frequently and moved further away from their previous home range center. Thus, hedgehogs avoided the core festival area, changed their behavioural budget and their activity patterns indicated stress.

In fragmented areas, hedgehogs increased home range size and movement speed without any significant effects on their behaviour budget or synchrony to environmental 24 h rhythm.

These results suggest that hedgehogs can adjust to permanent disturbances such as fragmentation. It was also interesting to see that disturbance affected females differently to males; males seemed to be more active in avoiding or coping with the environmental changes. These findings show that urban hedgehog populations can be resilient to transient as well as permanent habitat changes. However, care should be taken when extrapolating our results to other urban environments.

### 4.1. Home Range Sizes

Consistent with our predictions, nightly home ranges decreased during the festival (A1b) and home range size was enlarged in highly fragmented areas (B1). Fragmentation increased the neglected area and thus increased the distances hedgehogs moved between resource patches. If hedgehogs in the highly fragmented area accessed one spacious food patch the size of home ranges did not vary between highly and low-fragmented areas [63].

Our home range sizes are consistent with earlier findings for urban areas [35] and in other hedgehog species in suburban habitats [64] but are smaller than in many studies [65,66,67,68,69]. However, these studies calculated 100% MCP from longer tracking periods, which commonly leads to bigger home range than 95% MCP of high-resolution daily data. Furthermore, home range size may not be a constant over time, as previous studies showed big differences in individuals which were compared between two consecutive years [35]. Our results are consistent with the general conclusion that males used larger areas than females [70]. In a study on hedgehogs in rural habitats, hedgehogs further away from settlements had higher energy expenditure, presumably because they had longer distances to cover [71]. The same study showed that hedgehogs may restrict their movements in the presence of predators such as badgers. It is, therefore, important to know whether the trade-off between the use of spatially fragmented areas (and concomitant high energy expenditure) and the risk of encountering predators is biased in favour of hedgehogs in closer proximity to settlements [71].

### 4.2. Movement Speed and Turning Angle

A comparative study on movement behaviour of mammals demonstrated a negative effect of anthropogenic disturbance on long distance displacements [72]. In key indicators of movement behaviour, our results were consistent with predictions that during transient disturbance, hedgehogs should increase search intensity, enlarge their turning angles and reduce speed (A2). Also, in the highly fragmented areas, hedgehogs moved at a higher speed and increased the number of (smaller) turning angles but we did not find the predicted lower search intensity than in low-fragmented habitat (B2).

In our study, we calculated mean speed based on GPS positions measured during both periods of active and inactive behaviours. We report similar values to the mean (average over sex and season) speed of Ethiopian hedgehogs of 0.039 m/s [70]. We identified differences between highly and low fragmented areas, showing that hedgehogs in highly fragmented areas moved on average by 20% faster, a substantial increase and relevant for total energy turnover [71]. In former studies, it was shown that hedgehogs move faster when passing bigger roads, which could mean that hedgehogs on tarmacked or concrete paths in the fragmented area in the Tierpark also increase their speed if they use them [73]. Higher speed in the highly fragmented habitat could also be a consequence of commuting between foraging patches, whereas slow movements in hedgehogs are often indicative of foraging [34,74]. These factors will increase the variance in search intensity in the highly fragmented habitat.

### 4.3. Centre of Nightly Home Range

Our results were consistent with the predictions regarding a shift of the centre of nightly home ranges during the transient disturbance (introduction (A1a)).

The detected spatial shift of the nightly used core area is a clear indicator that hedgehogs avoided the core festival area. As the open park areas, where the local hedgehogs usually foraged, was blocked by festival visitors or, during the construction work, by workers, hedgehogs stayed for longer periods, and sometimes the entire night, in areas on the edge of the park. Or they did not leave the bushes where their nest where located. The observed sudden shift of the home range centroids which is unusual for the season and linked to the timing of the festival is a clear sign of avoidance. Such a response was also shown by hedgehogs living in cultivated land as a response to dramatic changes in resource quality [75]. In a similar semi-experimental approach, koalas (*Phascolarctos cinereus*) were tracked before and during a music festival and changed their home ranges in a manner similar to the hedgehogs in Berlin during the festival [76,77]. Hence, it is important to facilitate avoidance behaviour by creating escape routes for wildlife during events such as music festivals.

### 4.4. Behaviour Parameter

Where and when an animal conducts certain behaviours are crucial for understanding habitat use as well as elucidating the response of wildlife to disturbances as changes in behavioural patterns can be the consequence of disturbance [57].

Our results were consistent with predictions that during transient disturbance, the general level of hedgehog activity will be reduced as vigilance behaviour should increase (A3). During the festival, we observed an increase in the frequency of curling-up or roll-up behaviour. Disturbance first induces an erection of spikes, and then is followed by a complete roll up if the animal is further stressed [19]. When analysing our data, we have to be careful about the possible misidentification between resting and rolled up behaviour. The huge 95% confidence interval in the behavioural parameters suggests a high variability in the behavioural response. Thus, visitors and noises of the festival as well as natural predators such as red foxes and badgers that roam in both highly and low-fragmented areas could cause curling-up behaviour. An alternative response to disturbance is that hedgehogs can run away from a disturbance or a predator, which in turn would lead to increased activity levels. Hedgehogs are known to move out of unfavourable habitats and thus could avoid disturbances if necessary [78].

Although the size of the area used in the highly fragmented habitat was increased, it was not clear whether female hedgehogs may have a higher energetic investment there [71] as they covered similar distances. In terms of behaviour parameters, there was no obvious general pattern as to how hedgehogs responded to different disturbances. Such high inter-individual behavioural variance in response to anthropogenic disturbance is an expression of behavioural flexibility and may contribute to successfully persist in challenging environments such as big cities [54]. However, behavioural flexibility also has the evolutionary cost that a chosen behavioural response may be inappropriate in that situation [79], and these are more likely if the anthropogenic disturbance is unequal to natural disturbances to which there may be an evolved response [41].

Rhythms in behaviours have evolved as adaptations to the environment and enable organisms to behave at the times most suited to their physiology or ecology. DFCs are calculated as a measure of harmony between behavioural rhythms and the most important environmental rhythm, the 24 h period. High DFCs are often found in healthy animals or those that are strongly diurnal or nocturnal [80]. Low DFCs indicate that the animal is weakly synchronized with the environmental rhythm, which can be an indicator of stressors or disease, but also parturition [57,81]. In our study, DFC clearly decreased during the transient disturbance in both sexes, while there was no change detected with regard to fragmentation. Interestingly, we found the highest DFC values in females in a highly fragmented habitat, suggesting that they had adjusted their biological rhythm to the environmental circumstances. DFC values of male hedgehogs in the highly fragmented area had a higher variation, i.e., displayed higher individual differences during this time of the year. This could be expected as male hedgehogs may vary in their behaviour during and shortly after the mating season. Some of the females in our study gave birth to hoglets during the study period, which could have influenced the DFC values in addition to the disturbances. To disentangle the effect of reproductive status (such as parturition, lactation) from transient disturbances, studies in semi-natural enclosures could give baseline data for an energetic and hormonal assessment and for evaluation of DFC values as a non-invasive tool to assess the stress response in wildlife [57].

### 4.5. Nesting Behaviour

Consistent with our predictions, male hedgehogs switched their nests more often and spent fewer days in the same nest during the transient disturbance of the festival (A4). In contrast, females used their nests longer, which might be related to the fact that some of them gave birth during the study period and thus were restricted to stay in their nests.

Nocturnal hedgehogs change their day nests regularly. Changing nests more frequently rather than reusing a previously built nest entails a higher expenditure of energy. In our study, we recorded much higher numbers of nest changes within a period of up to 40 days than in Irish populations of rural hedgehogs (our study mean values 7.5 in males, 4.9 in females vs. 2.5 in both sexes [37]) but similar ranges to hedgehogs on a golf course in the suburbs of London (males 5–10 (our study) vs. 2–15, females 2–6 (our study) vs. 2–6 [36]). Our females used fewer nesting sites than males, similar to the London study [36]. There are currently no other survival analyses of nesting studies available. With access to data with day to day records it should be possible to find a baseline for European hedgehogs.

### 4.6. Limitations and Outlook

Overall, we could show that males responded stronger to the transient disturbance, although caring for hoglets may have diminished the response of some females to the transient disturbance of the festival. In contrast, female hedgehogs responded stronger to habitat fragmentation, possibly a consequence of their higher energy requirements for lactation. Details on individual differences in coping strategies as a function of reproductive status or other aspects of individual life histories could help predict responses to specific disturbances. Individual behavioural flexibility is, therefore, an important issue for further studies [17,82].

In an anthropogenically changed environment, some species maybe have already adjusted their behaviour to suboptimal habitats to avoid direct conflicts with people [83,84]. Thus, hedgehogs living in urban areas may have already adjusted their behaviour to urban environments. The responses to the transient disturbance of the festival clearly indicate that this constitutes a habitat change which would worsen the situation.

Female hedgehogs in the highly fragmented areas increased their speed and the size of their home ranges, which might either indicate higher energy requirements satisfied by abundant resources, or energetically more efficient locomotion [7]. The right combination of state-of-the-art technologies will enable us in future to study such subtle coping strategies. Nowadays, advanced technologies offer a level of detail in behavioural studies of free-ranging wild animals that has previously been impossible. This will improve our understanding of the role of behavioural mechanisms in ecological and evolutionary processes [85], provided care is taken to combine on the ground close population monitoring with advanced remote technologies [86,87].

## Figures and Tables

**Figure 1 animals-10-02109-f001:**
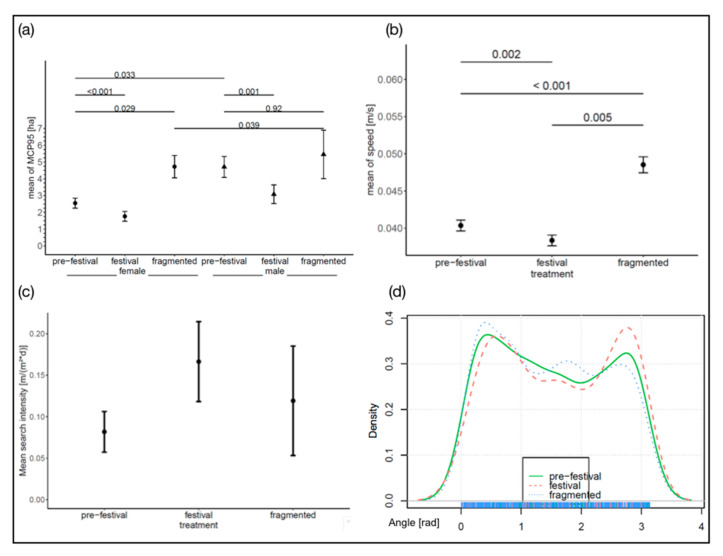
Results from the GPS data analyses. (**a**) MCP 95 measured per treatment and sex per day (dots/triangles indicate mean values, whiskers are confidence interval); (**b**) mean speed (m/s) for each treatment over both sexes (dots indicate mean values, whiskers are confidence interval); (**c**) mean search intensity (m/(m^2^ × d)) for each treatment over both sexes (dots indicate mean values, whiskers are confidence interval); (**d**) Distribution of absolute turning angles of one subsample that were tested.

**Figure 2 animals-10-02109-f002:**
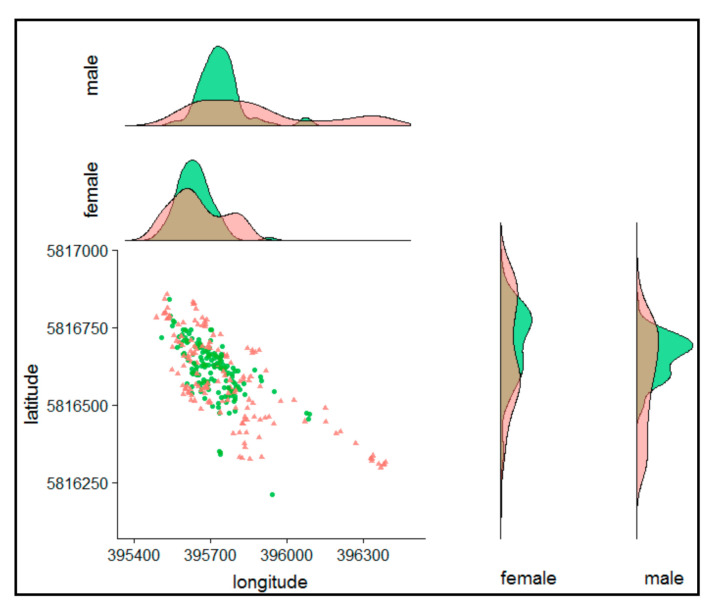
Centroid distribution. Circle/green = pre-festival period, triangle/red = festival period; density plots show the total of kernel density estimates (kde50) in the same range.

**Figure 3 animals-10-02109-f003:**
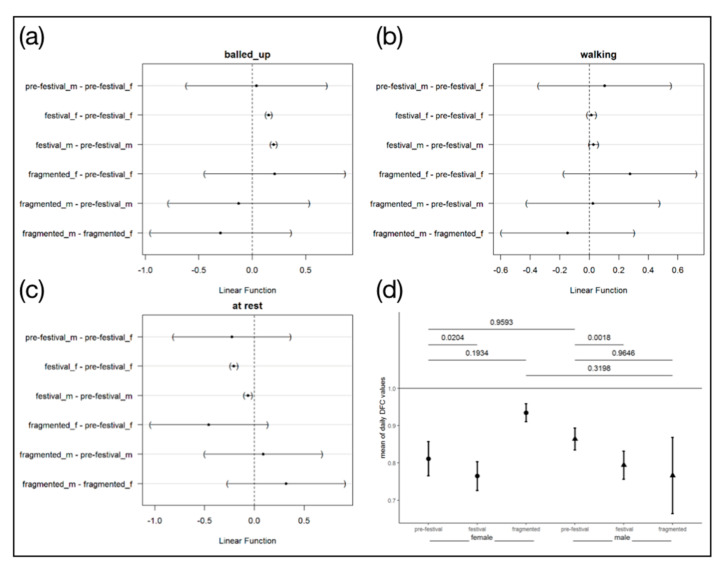
Behaviour parameters in the context of transient and fragmentation habitat changes. Mean values with 95% confidence interval for differences in behaviour parameter counts of balling up (**a**), locomotion (**b**) and immobile behaviour (**c**) of female (f) and male (m) hedgehogs for the different study phases. The difference represents changes in behaviour event counts. Negative values represent a decrease in the behaviour counts while positive values represent an increase. We considered all differences to be significant where the confidence interval does not include 0 (dashed line); (**d**) Mean degrees of functional coupling for each study period and sex (1 means maximal synchrony animal’s between behaviour and the environmental 24 h period, 0 means no synchrony), whiskers = 95% family-wise confident intervals.

**Figure 4 animals-10-02109-f004:**
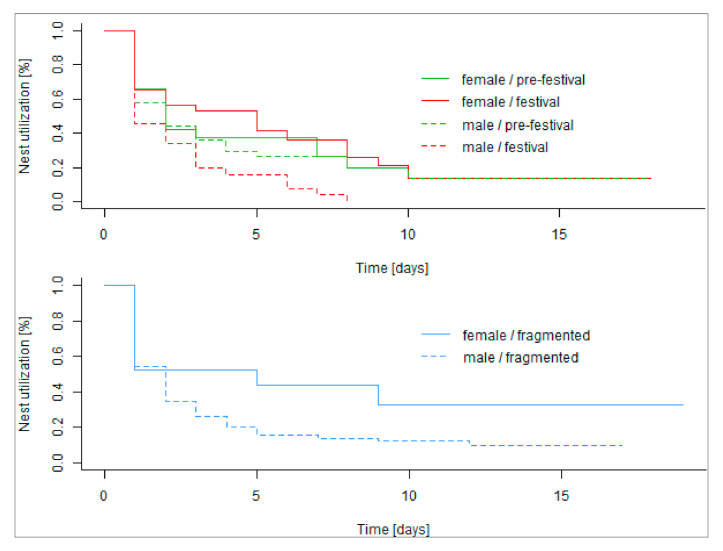
Nest utilization probability for pre-festival (green) and festival (red) phase of nine males (dashed line) and eight females (solid line) and for highly fragmented area (blue) with shortened *x*-axis for comparison.

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
