# Peer review of "Urban Hedgehog Behavioural Responses to Temporary Habitat Disturbance versus Permanent Fragmentation"

_animals, 2020, doi:10.3390/ani10112109_

Round 1

Reviewer 1 Report

The authors have implemented most of the changes suggested in the previous round of review, or provided an explanation where this was not done. The message of the manuscript is consequently clearer and I only have a few minor suggestions related to the changes made.

L44: I think you can make the closing sentence a little more general, something like “Our results are relevant for the welfare and conservation measure of urban wildlife and highlights the importance of allowing wildlife to avoid urban music festivals by facilitating avoidance behaviours.”

L226: Please state in the text the number of locations removed.

L226: is there a reference to the preliminary study – or do you mean a preliminary analysis?

L237: you write “used are adults means that they are born at least the year before the investigations”, I suggest you change to “were adults, meaning that they were born at a year before the study”

L239: Following your response to my previous comments about obtaining p-values for the fixed effects, it now becomes clear to me that you performed the anova on the fitted model object. This was not originally clear to me in the text. Can you please make this more explicit, something like “To obtain p-values of the mixed-effect model, an analysis-of-variance was performed on the fitted model using the Anova function from package car (citation). This applies a Wald chi-square test which compares the coefficient with the estimated standard error. Subsequently, difference between groups were tested using general linear hypotheses in the R package multcomp (citation), and the groups that were compared were: xxxx”

L273: Please state in the number of days removed (i.e. not just in the response).

L325: Thank you for adding the weather data, it certainly helps aid the comparison between the two years and reduces any uncertainties due to external factors. However, at present the description of the weather results stand alone and there is no indication in the methods why you measure the weather. You could also include some information in the discussion that you have in your responses about the reliability of the comparisons in your study design.

L434: You are too harsh on yourselves here, I think it is okay to put something like “care should be taken when extrapolating our results to other urban environments”.

L488: “it is important to facilitate”

Supplementary R code: please include a call at the beginning of the script that calls in all the libraries you used. Some functions, like Anova, can be found in different packages. At some points you specifically have car:Anova but this is not always, and for example in other functions like glht. This helps any readers that wish to follow the same type of analysis at a later stage. You should also state which version of R you are using (and in the manuscript please include a citation to R Core Team. 

Author Response

The authors have implemented most of the changes suggested in the previous round of review, or provided an explanation where this was not done. The message of the manuscript is consequently clearer and I only have a few minor suggestions related to the changes made.

L44: I think you can make the closing sentence a little more general, something like “Our results are relevant for the welfare and conservation measure of urban wildlife and highlights the importance of allowing wildlife to avoid urban music festivals by facilitating avoidance behaviours.”

Response: We changed as proposed.

L226: Please state in the text the number of locations removed.

Response: We added this information in Chapter “3. Results” and Table S1.

L226: is there a reference to the preliminary study – or do you mean a preliminary analysis?

Response: The assumed preliminary analysis is correct. We adjusted the text.

L237: you write “used are adults means that they are born at least the year before the investigations”, I suggest you change to “were adults, meaning that they were born at a year before the study”

Response: We changed as requested.

L239: Following your response to my previous comments about obtaining p-values for the fixed effects, it now becomes clear to me that you performed the anova on the fitted model object. This was not originally clear to me in the text. Can you please make this more explicit, something like “To obtain p-values of the mixed-effect model, an analysis-of-variance was performed on the fitted model using the Anova function from package car (citation). This applies a Wald chi-square test which compares the coefficient with the estimated standard error. Subsequently, difference between groups were tested using general linear hypotheses in the R package multcomp (citation), and the groups that were compared were: xxxx”

Response: We adjusted the explanation and added information about the type of ANOVA used as well as the way we applied the group comparison. Now we write: “To obtain p-values of the mixed-effect model, an ANOVA (Type II) was performed on the fitted model using the ‘Anova’ function from R-package ‘car’ applying a Wald chi-square tests [47]. Subsequently, differences between groups of the fixed-effects of the fitted model were tested using the R package ‘multcomp’ [48]. We compared the groups: … “

L273: Please state in the number of days removed (i.e. not just in the response).

Response: We added this information in Chapter “3. Results”.

L325: Thank you for adding the weather data, it certainly helps aid the comparison between the two years and reduces any uncertainties due to external factors. However, at present the description of the weather results stand alone and there is no indication in the methods why you measure the weather. You could also include some information in the discussion that you have in your responses about the reliability of the comparisons in your study design.

Response: We changed as requested (see chapter “4. Discussion”).

L434: You are too harsh on yourselves here, I think it is okay to put something like “care should be taken when extrapolating our results to other urban environments”.

Response: We changed as requested.

L488: “it is important to facilitate”

Supplementary R code: please include a call at the beginning of the script that calls in all the libraries you used. Some functions, like Anova, can be found in different packages. At some points you specifically have car:Anova but this is not always, and for example in other functions like glht. This helps any readers that wish to follow the same type of analysis at a later stage. You should also state which version of R you are using (and in the manuscript please include a citation to R Core Team. 

Response: All requested information is now included in the Supplementary Table 3 under ‘Session Info’ at the beginning of the script. It shows all used packages as well as the R-software version.
In the attached html-notebooks, warnings messages are included, hence masking of functions would be indicated in the notebook. If a specific/not default function was used like “car:Anova” the package was included in the function call. ‘glht’ is a function of the ‘multcomp’ package and not used in another package.

The requested R Core Team citation was already included as citation number 43.

Reviewer 2 Report

This manuscript is much improved and the authors have addressed all my previous comments or provided a good rationale for not doing so where applicable.

Author Response

There were no further comments or questions, thus we do not have any Responses.

This manuscript is a resubmission of an earlier submission. The following is a list of the peer review reports and author responses from that submission.

Round 1

Reviewer 1 Report

The authors present a fascinating analysis of the response of 8 hedgehogs (4 males and 4 females) to a music festival suddenly appearing in their home range.  They also contrasted the behaviour of these 8 hedgehogs to that of 8 others the following year in a different site.  I considered the latter comparison to be weak with the obvious problem of lack of site replication – besides being done in different years in different places, the two places likely differ in ways besides the habitat fragmentation difference that the authors highlight.  For example, the Tierpark is closed at night vs Treptower is not.  The comparison which is meant to study effects of habitat fragmentation seems scarcely worth including in this paper.

For studying the music festival effect, having only 8 animals is minimal, less than ideal, but acceptable. 

For the analysis of changes in behaviour before vs during a music festival, the results are interesting, but would be much more insightful if the patterns were explicitly and quantitatively coupled with habitat information.  For example, the authors note that the animals shifted from the open grass meadow to the edge of the park – sometimes not leaving the bushes.  While this sounds like a plausible observation, it is ‘just an anecdote’.  If, however, the authors had some mapping of habitat types and contrasted each animal’s space use and movement before vs during as mapped on an actual landscape with habitat types, and, for example, the location of music festival activities, a story like moving from the meadow to the bushes could go beyond anecdotal.  Similarly, male hedgehogs moved their nests more often during the festival.  Including information on where their nests were located relative to festival activities would be informative.  Beyond this sense of lost opportunity to do more with the data, I only had minor comments that I will list in order of appearance in the paper.

Please report the best estimate of how many hedgehogs were in the festival park, including whether there were likely animals already living in areas along the edge of the park.  That is, did the 8 focal animals probably need to respond to numerous other hedgehogs in their response to the music festival?

The overall experimental design is simple enough that figure 1 is not really necessary.

The authors seemed TOO careful about dropping data points.  If there were ANY missing GPS readings in an entire day, the day was dropped.  This seems too extreme.  How many days had to be dropped by this criterion?  Similarly, if any accelerometer data per burst was missing, the burst was removed.  This again seems rather extreme.  How many bursts had to be dropped? 

Often, GPS readings that are clearly incorrect need to be filtered out.  There are R packages for doing that.  Please describe methods for doing this. 

Line 236 – please tell us what package in R was used for the linear mixed model analyses.

Line 245.  Movement speed was calculated for all time intervals between 2 consecutive GPS positions.  Did the animals often move a small enough distance in 10 min that the change in GPS position could be heavily affected by GPS error?  More broadly, what was the typical GPS error (e.g., +/- 5 or 6 m?) and how was that error variance accounted for in the analyses? 

Line 264- It was unclear why turning angles that could be calculated for every change in GPS position were not analyzed in the same was as movement speed, but were instead randomly sampled with a permutation approach and compared only at the treatment level as opposed to via a linear mixed effects model.

Line 282: It was unclear what time period was used for a data point.  I assume that an accelerometer burst was taken every 10 min.  Were these bursts pooled to yield a proportion of behaviour exhibited of a given type per day (per individual)? 

Figure 2: on the lat/long map – it might be good to show males vs females in different colors.  It appears that most of the animals that shifted substantially to the south and east were males.  What habitat type was this new previously unused area?  Also, given that this represents only 8 animals, it might be useful to show the overall map of centroids for each both before and after, as opposed to showing every night for every animal with no way for the reader to identify what individual animals are doing.

Figure 3 – the axis labels are too small to be readable. 

Line 340 says that HR sizes were 70-80 ha in fragmented habitat, while figure 3 seems to show only about 5 ha.  Which is correct?  Is the smaller value per day, while the larger number is for the entire study?  The variation metrics in the text are very small (only about 1% of the mean), while the ‘confidence intervals’ in the figure are proportionally larger.  Please clarify what these metrics and CI are – SD, SE, 95% CI?

Line 345: Are these movement speeds only including 10 min periods when an animal clearly moved, or does this include zeroes when animals were resting or balled up?  If it includes zeroes, what proportion of the time where they resting, and if it is a large proportion, the authors should probably use zero-inflated analyses.

Line 349: type – ‘search’ not ‘reach’.

The effect sizes in figure 4 allow us to see what comparisons were significantly different from no difference, but please tell the reader some basics – e.g., what proportion of time did males vs females in different treatments spend resting vs walking vs balled up?

Author Response

We are extremely grateful for the time and effort of the reviewers in providing peer review of our manuscript and the time afforded to us by the editorial team. The feedback has permitted us to improve our manuscript.

Please see the attachment for the detailed response.

Reviewer 2 Report

Barthel et al. investigate how two types of disturbance, namely a music festival (temporary) and habitat fragmentation (permanent) influence the movement and behaviour of hedgehogs in urban park habitats. The authors have performed a number of different analyses to compare across three treatments which include a control (unfragemented/pre-festival), a festival (unfragmented) and a fragmented (no festival) area. They show how movements become heavily restricted due to the festival and used areas shift as well. Hedgehogs in fragmented areas utilised larger areas and travelled at higher speeds. The results are important for understanding the impacts of disturbance on wildlife, and in this instance, how impacts already experienced by hedgehogs in urban areas can be heightened through intense periods of human activity. The article therefore makes an important contribution, but unfortunately at present I don’t believe the “fragmented” area is truly comparable and it brings a number of uncertainties into the analysis. I fully appreciate the authors intentions of comparing permanent and temporary disturbances, and such a topic may have broader appeal, but I do not believe the data allow for an accurate comparison and outline my concerns below.

Major comments

My principal concern is with the Tierpark (fragmented habitat). A major methodological hurdle is that the GPS data for this study site was collected in a different year. No thought is given to how environmental conditions and food availability may vary between these two years, and consequently how this may affect movements. As the authors themselves note in the discussion (L432), “home range size may not be a constant over time, as previous studies showed big differences in individuals which were compared between two consecutive years”. Consequently, it is near impossible to tease apart differences in environmental conditions and food from behavioural differences at the two sites. I know this is not the kind of comment you’d like to receive, because it is impossible to go back and change how the data was collected. As I outline below, I am also concerned about the very general way that a habitat is described as “fragmented”, and as such, I would actually suggest that you streamline the manuscript to show the large impacts that a temporary (but annual?) disturbance can have on hedgehogs – especially as this disturbance occurs during the reproduction season for hedgehogs (you mention females had hoglets in the discussion). Consequently the message and analyses become much simpler to explain – and the comparison of the control and festival already form the core message of the manuscript – with generally less attention devoted to the fragmented area.

As mentioned above, your use of “fragmented” habitat is quite general, is never quantified and it seems to be largely due to zoo enclosures and paved paths. However, your non-fragmented habitat is itself an urban park, with many foot paths crossing the park, there is a major road that cuts the park in two, with cycle paths on either side of the road, and various buildings (including the huge war memorial) in the centre of the park. I would therefore argue that this is not unfragmented. Perhaps it is less fragmented, but this is not quantified. The GPS data is never displayed either so it is impossible to judge how hedgehogs may be moving through the landscape. One could also argue that the movement patterns you observe are not due to fragmentation, but instead that Treptower is exposed to continuous disturbance because it is accessible 24/7. This could explain for example the much smaller home ranges and speed (both control and festival) than Tierpark, because hedgehogs have to regulate their movements in response to human presence (this is also in line with an article you cite by Tucker et al.). Human presence subsequently increases even further with the festival. Hence, given that data was collected over two different years, and the uncertainties surrounding fragmentation, my suggestion is to focus the manuscript on Treptower only. Perhaps it is possible to address the concerns I raise above and revise the analyses but I imagine this will be difficult to achieve to accurately compare the two sites.

Line comments

L29: This conclusion should be taken in context of your study area though. It was conducted in city parks where there are likely no predators (badgers). Fragmented habitats could have much larger effects where predators are present and it becomes a risk (thus stressful) to move between habitat fragments.

L45: The question is how these “behavioural” effects may influence the life history of the species (survival/reproduction)

L70: Experienced is a strong word. This suggests that the habitat fragmentation occurred during the study period. In actual fact, the habitat (a Berlin park) has likely had its structure for a long time and thus the hedgehogs simply live in a fragmented habitat.

L83: second time the latin name of the hedgehog is used (see L69)

L114: these fitness consequences are never explored however. In the discussion you repeatedly return to the fact that several females had hoglets in both the fragmented and transient groups. Was there any difference in reproductive performance between the two groups?

L116: “Leave an unfavourable habitat” – I would imagine this response with or without a transient disturbance. I presume what you mean is that a habitat becomes unfavourable because of the transient disturbance – so I would be more explicit in your prediction.

L121: We predict rather than we propose?

L130: This line only refers to urban areas though. Hof et al. (2012) https://doi.org/10.1016/j.anbehav.2012.01.042 show that the movement also depends on presence of predators. In agricultural areas, where badgers were present, hedgehogs stayed much closer to linear features but where absent, they would venture into fields and thus not follow linear features. I therefore feel this statement is too much of a generalisation – you are describing a pattern for hedgehogs which are already disturbed. Urban areas can also be considered landscapes of fear given the threat of humans (and their pets), which would explain the use of linear features in urban habitats. You could perhaps update the sentence to state that “In urban areas, or in areas with predation risk, hedgehog movements are strongly associated with linear structures (refs), ….

L137: What do you predict though

L144 – L146: A concern in a comparative analysis like this, is how comparable were the summers of 2016 and 2017. Were temperature and rainfall similar and how might this impact food availability? All these patterns may explain difference in home range size but they are not considered in your analysis. For example, a warmer drier summer will make earthworms harder to find and therefore they need to roam further to meet their energy requirements. This alternative hypothesis could explain the results of Tierpark, instead of fragmented versus unfragmented, but how comparable the environmental conditions were are not investigate nor discussed.

L147: Grass swords would be really cool, but I suspect you mean grass swards.

L144 – L155: It is a fine line you draw between fragmented and unfragmented habitats. Treptower park also has a major road dissecting the park (with cycle paths too) which Tier park does not. In addition, Treptower has numerous foot paths (which may actually pose a greater obstacle because the park can be accessed 24/7) and various structures within the park. Tier Park on the other hand is quite unique with the zoo enclosures (are they truly inaccessible to hedgehogs?) but the complete absence of human disturbance at night. As per my major comment above, I think you need to either reconsider how you compare the parks, or focus on Treptower park because at present I see too many confounding variables that get lumped into a “fragmented” explanation for Tierpark.  

L166: As per above, I find it presumptive to argue that Treptower park is unfragmented and undisturbed. As you state earlier in the text, it is accessible 24/7 meaning there is a potentially continuous background level of disturbance and the various paths/roads/structures do not really count as unfragmented.

It would be informative to provide the GPS locations of the hedgehogs, or at least overlay the centroids shown in Figure 2 with the habitat structure of the parks shown in Figure 1.

L193: At what frequency were the five bursts for GPS points? What did you do with the five points (assuming they were all clustered around the 10 min interval). Did you use all 5 points, take a mean or median or something else?

L227: How many days were removed from the analysis?

L227: In addition, please clarify your data cleansing. 1440 points suggest you were using the behavioural data which was collected every minute (24 hours x 60 minutes = 1440). However, the accelerometer data has no GPS information attached. I would expect 12 hours (7pm – 7am) x 6 locations per hour (i.e. every 10 minutes) = 72 GPS positions per day (or 360 if you expect all five bursts). Or did you apply different cleansing protocols for behaviorual and GPS data (note that this section is titled “GPS data”)

 L229: This sentence contradicts L193 where you state that positions were only recorded at night (7pm – 7am)

L236: What package did you use to fit the model. Given that you include a varPower() function, I presume it was not lme4 which you use for the movement speed.

L241 and L250: From what I gather from your explanation, the P-values you report are whether a model provides a significantly better fit than another model (for example, if a model includes sex – is it significantly better than a model that excludes sex). However, this approach doesn’t actually provide the significance of the variables (fixed effects) themselves. As standard, lme4 tends not to report p-values (there is a large discussion on the forums). However, numerous methods exist to extract the significance of fixed effects. A good starts is the package lmerTest https://doi.org/10.18637/jss.v082.i13

L258: Working with both latitude and longitude complicates the aims of the analysis in my opinion. It appears that you already calculate a mean centroid for the pre-festival phase. There are numerous R packages that then allow you to estimate the distance from the nightly centroids during the festival phase to the mean centroid in the pre-festival phase. This could even be done manually using for example the haversine formula. Such an approach would make the results easier to interpret that absolute values of longitude and latitudes that have been normalised.

L262: Please clarify your fixed and random effects in the mixed effects model.

L282: You mean a mixed effects model (or a generalised linear mixed effects) – but in any case it wasn’t only a general linear model.

Methods general: Please provide a (supplementary) table providing information about the hedgehogs included in the analysis. You perform a number of filtering steps, with data ranging from 3 nights to 41 nights, so it is difficult to grasp how the 16 individuals contribute date to your three treatments and the various sub-analyses (HR, Behaviour etc.).

Results general: You perform a number of different analyses, and in these instances it usually helps the reader if the results are presented in the same order that the methods are described (in this case HR, speed, centroid, turning angle etc.)

Results general: Most results are fully explained in the text. It would be much easier to interpret these in a table. In addition, you never present the actual results of the mixed effect models (as far as I can tell) and only present the results of the model comparisons. Presenting the model results would be much more informative because then the reader can see the average (e.g. home range size) via the Intercept whilst the coefficients show the influence of sex and/or treatment.

L432 – L434: This sentence also supports why I feel comparisons between Tier and Treptower are complicated given the different years of data collection.

L438 – L441: Predators are not described in your present study, but I presume no predators are found (i.e. badgers). Foxes are not really a threat to (adult) hedgehogs in general.

L450: This sentence slightly stretches the results, or it needs to be placed in better context. Lat/long moved by 30/35m for females, but I imagine the nightly home range easily covers an area much larger than this.

L450: Further to above, rather than comparing distances between centroids, could you also detect a shift in nesting locations between pre-festival and festival?

Discussion general: a large emphasis is placed on how females with young may influence the results. This is indeed an important confounding factor because females with young will likely not leave their nests, they may either stay close to the nest but may also need to move over longer distances to find forage for their young and hence can have contrasting results. Did all females have young, or is there a way to compare your results between females with and without hoglets to understand a little more of how the reproductive status may influence your results.

Figure 2: Could you instead colour the points per individual. It is not entirely necessary to use both a shape and colour for a single comparison, so you could instead use colour for individuals and shape for pre-festival and festival. Could you also add (using a star for example) the mean centroid used for each individual in the analysis.

Author Response

We are extremely grateful for the time and effort of the reviewers in providing peer review of our manuscript and the time afforded to us by the editorial team. The feedback has permitted us to improve our manuscript.

Please see the attachment for detailed response.

Reviewer 3 Report

This is an interesting, well executed study. There are study design issues and the ms needs to be more concise with more clarity and restructuring in places, as well as improvement of figure presentation. However the concepts are well presented and discussed and topic is under-studied so is a useful contribution in my view.

Review of Barthel et al.

General comments

The manuscript by Barthel et al reports on research investigating hedgehog spatial behaviour in response to long-term habitat fragmentation and short-term anthropogenic disturbance in two urban greenspace areas (parks) in Berlin, Germany. The concept is really interesting because the short-term disturbance is a music festival, and to my knowledge the effects on wildlife of such events are rarely quantified.

Strengths:

The MS is well written, and the study design carefully considers spatial and temporal controls. The data analysis is comprehensive and appropriate, and the topic is important. Hypotheses are generally clearly stated and meaningful and the theoretical framework is really good.

Weaknesses:

The study design is unbalanced with no replication; only one site represents each of: 1) temporary disturbance to a non-fragmented environment, 2) no disturbance to a fragmented environment, and 3) no disturbance to a non-fragmented environment (with the latter being the same as 1 but as temporal control). This means that the results can not be extrapolated beyond the context in which they were collected.

The MS could be more concise and condensed to deliver a punchier message. The figures need improvement, both in terms of clarity of meaning and aesthetics. There is some repetition and minor issues with structure, which I attempt to address in the comments below.   

Specific comments (not an exhaustive list):

  1. The title is too long and not very clear, in my opinion. I suggest rewording to something like ‘Urban hedgehog behavioural responses to temporary habitat disturbance versus permanent fragmentation’
  2. First sentence of simple summary oddly worded. Suggest ‘environmental disturbances’ rather than disturbances to their environment.
  3. Line 24 – should it not be conservation action or prescriptions or management rather than concepts?
  4. I think the sex differences in hog response should be in the abstracts
  5. 36 – check the quotation marks. Also – could this not be put more simply? Why not just say ‘using temporal and spatial controls?’
  6. 43 - Comma after which
  7. 60-63 need better referencing
  8. 67-8. This section should really be at the end of the Introduction, no? I understand why it is here in this case – i.e. to explain the context, after which the section about festivals can follow. However, I think you can restructure the Introduction with a bit of reworking of the text so that this section comes at the end.
  9. This sentence is odd. Why not rephrase to something like ‘Increasing fragmentation and decreasing hedgehog densities may cause negative genetic effects associated with isolated populations’.
  10. ‘Parks’ is a bit vague. Can you say urban green spaces such as public parks?
  11. What mosaic-like structure? This sentence comes out of nowhere, so you need to provide a link to it and/or restructure it to show that urban parks represent patches in a matrix of varying quality (If indeed that is what you want to say!).
  12. 98 needs a reference
  13. 99 can you use different words to re-evaluate and consider? They are not wrong per se but they suggest a cognitive process that the hedgehog in all probability doesn’t really undergo!
  14. This sentence is partially repeated from line 66 paragraph. I would either not mention the music festival until this point or introduce the Tierpark context earlier. As it is the Intro gives the reader information in a piecemeal fashion, which is a bit frustrating.
  15. Full stop (period) needed after ‘degrees’. Also, I suggest rewording this sentence by replacing ‘particularly’ with ‘where disturbances are experienced’ otherwise it states the obvious i.e. reproductive seasons affect fitness!
  16. 118 this sentence should be in Methods
  17. I think the lead-in to the hypotheses should refer (at least briefly) to concepts regarding contractionism versus expansionism in territoriality, e.g. as set out in the Resource Dispersion Hypothesis (Macdonald 1983, updated reference Macdonald and Johnson 2015). Or some discussion of why home ranges in rich, aggregated habitats should be smaller.
  18. 124 suggest ‘greater turning angles’ to be consistent with line 136. Also, faster speed on line 136 to reflect the slower speed predicted on line 124.
  19. Did you consider including number of turns as well as turning angle? The latter presumably indicates change in direction (e.g. 360° is a large angle that indicates a complete turn = total change I direction but doesn’t show whether they turn frequently).
  20. You need to say what size the different parks were
  21. I find Figure 1 not very intuitive or clear. What do the male and female signs mean in relation to study design? There is no reference to unfragmented habitats or temporary/permanent.
  22. 180 what does ACC stand for? These are not very commonly used, unlike GPS, so I think you need to spell out in full and explain a bit more here. For consistency also spell out GPS.
  23. Bursts of 5 points. This needs explaining as normally you only get one point from each time interval from GPS collars, even if there are 5 satellites that pick up and triangulate each point.
  24. 199 replace chapter with section.
  25. What was the rationale for 5 nests as a cutoff?
  26. 5 Analysis use the phrases temporal and spatial control
  27. 228 what is the rationale for omitting animals with fewer than 1430 fixes? This seems to me a waste of a potentially good dataset and needless loss of sample size! Why nor control for for daylength/different numbers of fixes in your analysis using a log-transformed offset in your statistical analyses instead?
  28. More justification of your choice of home range metrics is needed. At the very least you need to say what method you used to select an appropriate smoothing parameter (bandwidth or h) you used.
  29. Why did you not measure and use age as a covariate? Presumably because it was difficult to determine age? If so say so.
  30. 256 explain the point of using the SPaMM package. Was it to counter auto-correlation?
  31. 251 I don’t think you need the phrase general linear hypotheses. I would delete it. Just to confirm – you used a General Linear Model not a Generalised Linear Model, yes? It’s just that GLM usually stands for the latter. Either would be appropriate it’s just to be clear.
  32. Rephrase 317 so the first sentence leads into the second. Delete ’we recorded their spatial behaviour’ as this is repetition/methods.
  33. 345 insert ‘the’ before fragmented/unfragmented
  34. 393 over significantly shorter time periods
  35. 407 delete ‘in terms of their behaviour’
  36. 479 elucidating and insert as after disturbances

Author Response

(The authors gave the same response as above.)
